# Preparation and Identification of Peptides with α-Glucosidase Inhibitory Activity from Shiitake Mushroom (*Lentinus edodes*) Protein

**DOI:** 10.3390/foods12132534

**Published:** 2023-06-29

**Authors:** Yu Zhang, Yu Chen, Xinyang Liu, Wei Wang, Junhong Wang, Xue Li, Suling Sun

**Affiliations:** 1Institute of Agro-Product Safety and Nutrition, Zhejiang Academy of Agricultural Sciences, Hangzhou 310021, China; zhangy_lk@zaas.ac.cn (Y.Z.);; 2School of Biological and Chemical Engineering, Zhejiang University of Science and Technology, Hangzhou 310012, China; 3Key Laboratory of Agricultural Product Information Traceability, Ministry of Agriculture and Rural Affairs of China, Hangzhou 310021, China; 4Zhejiang Provincial Key Laboratory of Food Safety, Hangzhou 310021, China; 5Collage of Agriculture and Animal Husbandry, Qinghai University, Xining 810016, China; 6College of Wine, North West Agriculture and Forestry University, Xi’an 712199, China

**Keywords:** shiitake mushroom, α-glucosidase inhibition peptides, identification, virtual screening

## Abstract

The shiitake mushroom is the most commonly cultivated edible mushroom in the world, and is rich in protein. This study aims to obtain the peptides with α-glucosidase inhibition activity from shiitake mushroom protein hydrolysate. The conditions of enzymatic hydrolysis of shiitake mushroom protein were optimized by response surface test. The results showed that the optimal conditions were as follows: the E/S was 3390 U/g, the solid–liquid ratio was 1:20, the hydrolysis temperature and time were 46 °C and 3.4 h, respectively, and the pH was 7. The active peptides were separated by gel filtration and identified by LC-MS/MS analysis and virtual screening. The results indicated that fourteen peptides were identified by LC-MS/MS. Among them, four new peptides (EGEPKLP, KDDLRSP, TPELKL, and LDYGKL) with the higher docking score were selected and chemically synthesized to verify their inhibition activity. The IC50 values of EGEPKLP, KDDLRSP, TPELKL, and LDYGKL for α-glucosidase inhibition activity ranged from 452 ± 36 μmol/L to 696 ± 39 μmol/L. The molecular docking results showed that the hydrogen bond and arene–cation bond were the two major interactions between four peptides and 2QMJ. The hydrogen bonds were crucial to the inhibition activity of α-glucosidase. The results indicate the potential of using the peptides from shiitake mushroom protein as functional food with α-glucosidase inhibition activity.

## 1. Introduction

Diabetes mellitus (DM) is a widespread metabolic disorder that affected 537 million adults globally in 2021 [1]. The number of diabetics is predicted to rise to 643 million by 2030 [2]. Two varieties of DM exist: type Ⅰ (T1DM) and type Ⅱ (T2DM). It is estimated that the number of people suffering from T2DM accounts for 90% of the total number of diabetics [3], and T2DM is characterized by hyperglycemia. Chronic damage and dysfunction of various tissues are caused by long-term hyperglycemia, such as cardiovascular disease, retinal damage, chronic kidney disease, and diabetic ketoacidosis [4,5,6].

The most effective treatment for T2DM is to reduce hyperglycemia, especially postprandial hyperglycemia, by slowing down the carbohydrate metabolism. The membrane-bound α-glucosidase, located in the small intestine’s epithelial mucosa, is a major digestive enzyme involved in carbohydrate metabolism. It can cleave the glycoside bonds in carbohydrate to free the glucose [7,8]. Therefore, inhibition of α-glucosidase activity has evolved to be an effective method for T2DM treatment because of its role in delaying the hydrolysis of carbohydrates to inhibit postprandial hyperglycemia [9,10]. Although the synthesized α-glucosidase inhibitors such as acarbose have been widely used for T2DM treatment, their long-term use leads to adverse effects [11,12]. Thus, more and more researchers are trying to look for α-glucosidase inhibitors from natural food as a complementary therapy for T2DM to ameliorate the adverse effects associated with synthetic drugs [13].

Recently, some plant or animal protein hydrolysates with α-glucosidase inhibition activity have been found, such as egg [14,15], bean [16,17], amaranth [18], quinoa [19,20], wheat gluten [21], and black cricket [22]. A variety of α-glucosidase inhibitory peptides were identified from Binglangjiang buffalo casein [10], walnut [23], egg protein [4,24,25], Spirulina platensis [26], Ginkgo biloba seed protein [27], camel whey protein [28], and Changii Radix hydrolysates [29].

Shiitake mushroom (*Lentinus edodes*) is the largest edible mushroom in China and is also the second most commonly cultivated edible mushroom in the world [30,31,32]. The yield of shiitake mushrooms in China has increased, but there are few processed products, which hinders the development of the shiitake mushroom industry. The shiitake mushroom is widely popular because of its nutritional value due to a variety of nutritional compounds including polysaccharides, protein, fiber, vitamins, and minerals [33,34,35]. At present, research on shiitake mushrooms mainly focuses on polysaccharides [35], and there are few studies on the development of shiitake mushroom protein. Shiitake mushroom is rich in protein, with a protein content of about 16–22% [34]. In addition, the shiitake mushroom protein contains abundant human essential and non-essential amino acids. Therefore, the shiitake mushroom, as one of the main sources of active peptides, has good prospects and may be crucial to the development of functional foods. The objective of this study is to optimize the preparation conditions of peptides and obtain the hydrolysate with the maximum inhibition activity of α-glucosidase. The peptides from the hydrolysate of shiitake mushroom protein were purified by gel chromatography, and the potential peptides with α-glucosidase inhibition activity were further identified by LC-MS/MS and virtual screening. Then, the relation between the structure of α-glucosidase inhibition peptides and activity was investigated.

## 2. Materials and Methods

### 2.1. Materials, Reagents and Instruments

Shiitake mushrooms were acquired from a farmers’ market. Neutral protease (50 U/mg), pepsin (1:30,000), alkaline protease (200 U/mg), acid protease (50 U/mg), papain (800 U/mg), and flavor protease (20 U/mg) were purchased from Qiyi Biotechnology (Shanghai, China) Co., Ltd. α-glucosidase (10 U/mg) and 4-nitrophenyl a-D-glucopyranoside (PNPG, 99% purity) were purchased from Sigma-Aldrich (St. Louis, MO, USA). Acetonitrile (HPLC grade, 99.9% purity) was purchased from Fisher Chemical (Hampton, NH, USA). Formic acid (LC-MS grade, 99% purity) was purchased from Thermo Scientific (Waltham, MA, USA). Other reagents were of analytical grade and commercially available.

An Thermo Easy-nanoLC system was connected to a Thermo Scientific QE mass spectrometer, USA Thermo Fisher Scientific. Other devices used were as follows: Analytical balance, Sartorius Scientific Instruments (Beijing) Co., Ltd. (Beijing, China). AMR-100 automatic enzyme label analyzer, Hangzhou Aosheng Instrument Co., Ltd. (Hangzhou, China). MTYK-MI805 pH meter, Beijing Zhonghui Tiancheng Technology Co., Ltd. (Beijing, China). HZQ-B Constant Temperature Culture Shaker, Suzhou Weier Laboratory Supplies Co., Ltd. (Suzhou, China). H-1650 high speed centrifuge, Changsha Xiangyi Centrifuge Instrument Co., Ltd. (Changsha, China). Freeze-drying Instrument, Shanghai Haozhuang Instrument Co., Ltd. (Shanghai, China).

### 2.2. Peptide Preparation and Condition Optimization

#### 2.2.1. Preparation of Peptides

Extraction of shiitake mushroom protein: A certain quantity of dried shiitake mushroom powders was dissolved in deionized water with the mass–volume ratio of 1:20. The solution pH was adjusted to 9.0 with 1 mol/L NaOH. The mixture was sonicated at 40 KHz and 50 °C for 60 min, then centrifugation was used to separate the supernatant for 15 min at a speed of 4000 rpm. The supernatant pH was adjusted to 3.5 by 1 mol/L HCl. Then, after centrifugation at 5000 rpm/min for 20 min, the precipitates were collected.

Hydrolysate preparation: Two grams of protein powder was dissolved in deionized water according to Section 2.2.2. A certain amount of protease was added according to Section 2.2.2, and mixed once the pH value was adjusted to an ideal level according to Section 2.2.2. The mixture was shaken at 150 rpm/min for a certain period time according to Section 2.2.2. The reaction was terminated in boiling water for 5 min. The pH was adjusted to 7.0 with 1 mol/L NaOH or 1 mol/L HCl according to the hydrolysis pH. Then the supernatant was collected after centrifugation at 5000 rpm/min for 20 min.

#### 2.2.2. Optimization of Single Factor Enzymatic Hydrolysis Conditions

With the E/S of 4000 U/g, the solid–liquid ratio of 20, the pH of 7.0, the hydrolysis temperature of 45 °C, and the hydrolysis time of 4 h as the basic fixed factors, one of the factors was changed and the single factor test was carried out. The gradient design of each factor was as follows: E/S (1000, 2000, 3000, 4000, 5000 U/g), solid–liquid ratio (15, 20, 25, 30, 35), pH (5, 6, 7, 8, 9), hydrolysis time (2, 3, 4, 5, 6), hydrolysis temperature (25, 35, 45, 55, 65 °C). The α-glucosidase inhibition activity of hydrolysate with different conditions was compared.

#### 2.2.3. Response Surface Methodology

The hydrolysis conditions of shiitake mushroom protein were improved using CCD. The inhibitory activity of α-glucosidase acted as the response value. According to the single factor experiment, E/S (A), temperature (B), and time (C) were selected to be optimized for response surface optimization. Table 1 displays the levels of the experimental factors.

### 2.3. Determination of α-Glucosidase Inhibition Activity

A quantity of 100 μL α-glucosidase solution (0.2 U/mL) was premixed with 50 μL sample solution in the enzyme label plate, and incubated at 25 °C for 10 min; 50 μL of PNPG (5 mmol/L) solution was added to start reactions, and incubation was continued for another 30 min at 37 °C. The reaction was terminated with 50 μL 0.67 mol/L Na_2_CO_3_ solution. The absorbance value was measured at 405 nm using an AMR-100 automatic enzyme label analyzer. The results were calculated using the following formula:Inhibitory rate (%) = ((A_control_ − A_sample_)/A_control_) × 100(1)

### 2.4. Purification of α-Glucosidase Inhibition Peptides

#### 2.4.1. Ultrafiltration

The hydrolysate solution was separated by ultrafiltration membranes of 1 k Da, 3 k Da, and 5 k Da. The fraction was named UF-I (>5 k Da), UF-II (3–5 k Da), UF-III (1–3 k Da), and UF-IV (<1 k Da), and collected separately.

#### 2.4.2. Gel Column Filtration

The fraction F-IV obtained by ultrafiltration was further separated by gel filtration according to Chen et al.’s method [36]. The sample was loaded onto a Sephadex G-10 gel filtration column (2.6 cm × 100 cm) after being dissolved in deionized water. Following that, deionized water was used to elute the peptides at a flow rate of 0.5 mL/min. The eluted fractions were monitored at 280 nm. The desired peak fractions were collected and lyophilized. The α-glucosidase inhibitory rates of elution peaks were determined.

### 2.5. Identification and Screening of the α-Glucosidase Peptide

The fraction with maximum activity obtained by gel filtration chromatography was analyzed by LC-MS/MS according to the method of Chen et al. [36].

The peptides were diluted in a 20 μL solution of 0.1% formic acid and 5% acetonitrile and separates with an Acclaim PepMap RPLC C18 (75 um i.d. × 150 mm,2 um, 100 Å, nanoViper) linked to a PepMap RSLC C18 in an LC-MS/MS system. The mobile phase was as follows: mobile phase A (0.1% formic acid) and mobile phase B (80% acetonitrile with 0.1% formic acid); the gradient elution was performed with a gradient of 3–99% B (0–34 min) and the flow rate was 500 nL/min. 

MS data were acquired over the range from 100 to 1550 *m*/*z* in ESI positive mode. Resolution was 120,000, AGC target was 4 × 10^5^, and maximum IT was 50 ms. MS/MS scanning conditions were as follows: resolution was 30,000, AGC target was 1 × 10^5^, maximum IT was 100 ms, TopN was 20, and NCE/stepped NCE was 32.

Peptide sequences were identified using PEAKS Studio X software in combination with a shiitake mushroom protein database search.

### 2.6. Molecular Docking Analysis

The semi-flexible molecular docking was performed between the identified peptides and the crystal structure of the human α-glucosidase (PDB code: 2QMJ) using MOE software according to Chen et al.’s method [36]. The crystal structure of 2QMJ (PDB DOI: 10.2210/pdb2QMJ/PDB) was downloaded from the PDB database. The structures of 14 peptides identified by LC-MS-MS were drawn using MOE2009 software. 

To obtain the receptor molecules required for docking, the water molecules from 2QMJ were eliminated, the molecules were protonated, and energy was minimized. The receptor pocket of the 2QMJ was used as the docking target for molecular docking, and the peptides were used as the ligand. Each coupling was performed 30 times. The peptide with a better docking effect was screened from the obtained results for further analysis.

### 2.7. Peptide Synthesis

The screened potential α-glucosidase inhibitory peptides were chemically synthesized by China Peptides Co., Ltd. (Shanghai, China). Fmoc solid-phase synthesis was used to obtain the raw peptide. Briefly, chlorine resin was swelled by dichloromethane (DCM). Fmoc-AA (amino acid)-OH, O-benzotriazol-1-yl-tetramethyluronium hexafluorophosphate (HBTU), and N,N-diisopropylethylamine (DIEA) were added to the drained resin for 30 min with a nitrogen bubble reaction to accomplish the condensation reaction. The resin was then washed with DMF to remove Fmoc. After being drained, cutting solution (trifluoroacetic acid (TFA)/H_2_O/1,2-ethanedithiol (EDT)/triisopropylsilane (Tis) = 95/1/2/2, *v*/*v*/*v*/*v*) was added to the resin to cut the peptide. The synthetic peptides were purified by HPLC. The molecular weight of the purified peptide was confirmed by mass spectrometry. 

### 2.8. Statistical Analysis

All experiments had three repetitions. Data are expressed as mean ± standard deviation. Statistical significance (Duncan’s test, *p* < 0.05) was analyzed by DPS. Response surface data were analyzed using Design-ExpertV8.0.6 software.

## 3. Results

### 3.1. Protease Selection

The α-glucosidase inhibition activity of shiitake mushroom protein hydrolysates obtained by hydrolysis with the acid protease, alkaline proteinase, neutral proteinase, flavor proteinase, papain, and pepsin was investigated. The results indicate that the enzyme specificity of the different proteases determines the activity characteristics of the hydrolysate. The α-glucosidase inhibition rates of the hydrolysates obtained with acid protease, alkaline proteinase, neutral proteinase, flavor proteinase, papain, and pepsin were 12.5 ± 0.31%, 14.8 ± 0.35%, 61.5 ± 1.6%, 33.2 ± 0.71%, 10.7 ± 0.28%, and 15.3 ± 0.37%, respectively. Compared with other hydrolysates, the neutral proteinase hydrolysate showed the maximum inhibitory rate against α-glucosidase at the same concentration. Therefore, neutral protease was chosen as the enzyme for hydrolysis.

### 3.2. Effects of Different Enzyme Concentrations on the Activity of Hydrolysates

The effect of enzyme concentration on the α-glucosidase inhibitory activity of shiitake mushroom protein hydrolysates was investigated. With the increase in the enzyme–substrate ratio (E/S), the α-glucosidase inhibitory rate increased first and then slightly decreased (Figure 1A). The α-glucosidase inhibitory rate of hydrolysate with E/S of 3000 U/g was significantly higher than that of 1000 U/g and 2000 U/g (*p* < 0.05). Therefore, 3000 U/g was selected as the appropriate enzyme concentration.

### 3.3. Effects of Different Temperature on the Activity of Hydrolysates

Temperature is a key influencing factor for enzymatic reaction. Increasing temperature can speed up the reaction rate, but excessive temperature can cause protease inactivation. The effect of different temperatures on the α-glucosidase inhibition activity of enzymatic hydrolysates was investigated. With the increase in temperature, the inhibitory rate of α-glucosidase first increased and then decreased. When the temperature reached 45 °C, the maximum inhibitory rate of α-glucosidase was attained (Figure 1B). Therefore, 45 °C was chosen as the appropriate condition. 

### 3.4. Effects of Different Substrate Concentrations on the Activity of Hydrolysates

The substrate concentration is a key parameter of enzymatic reaction. When the solid–liquid ratio was 15, the inhibitory rate of enzymatic hydrolysates was lower. When the solid–liquid ratio was 20, the α-glucosidase inhibitory rate of enzymatic hydrolysates increased and then decreased slightly (Figure 1C). It may be that when the solid–liquid ratio was 15, the protein concentration was too high, which affected the fluidity of the system and the contact surface between the enzyme and the substrate. When the solid-liquid ratio reached 20, the hydrolysate had good fluidity. Therefore, the solid–liquid ratio of 20 was chosen as the appropriate condition for enzymatic hydrolysis.

### 3.5. Effects of Different Enzymolysis Time on the Activity of Hydrolysates

The inhibition activity against α-glucosidase was significantly influenced by the enzymolysis time. With the increase in enzymolysis time, the inhibitory rate of α-glucosidase first increased and reached the highest level at 3 h, and then decreased (Figure 1D), which may be because some active peptides were further hydrolyzed with the increase in time. Therefore, 3 h was chosen as the appropriate condition for enzymatic hydrolysis.

### 3.6. Effects of Different pH on the Activity of Hydrolysates

With the increase in pH, α-glucosidase inhibitory rates of hydrolysates rose first and then decreased. When pH was 7, the inhibitory rate of α-glucosidase reached the highest level (Figure 1E). There were no significant differences in the α-glucosidase inhibitory rates among pH 6.0, pH 7.0, and pH 8.0. Therefore, pH 7 was selected as the appropriate condition.

### 3.7. Response Surface Analysis

In the actual process of enzymolysis, various factors may interact with each other. To further investigate the significance of the influence of various factors on the target value, E/S, temperature, and time were taken as independent variables, and α-glucosidase inhibitory rate as the response value, and a CCD experiment was conducted. The results are shown in Table 2. The quadratic polynomial regression equation of α-glucosidase inhibitory rate and E/S (A), enzymolysis temperature (B), and enzymolysis time (C) was obtained as follows:Y = 63.86 + 3.17A + 0.87B + 3.32C − 0.55AB − 0.87AC + 1.10BC − 3.49A^2^ − 4.89B^2^ −3.76C^2^(2)

According to the variance analysis results (Table 3), the regression model has a high level of significance (*p* < 0.0001). The R^2^ was 0.9877, which indicates that the test value and the fitting value had a high correlation. The regression equation model fitted the data well because the *p* value of the lack of fit was not significant. Consequently, it is possible to forecast the test results using the model. It can be seen from Table 3 and Figure 2 that there were significant interactions between temperature and time, and E/S and time.

The optimal conditions predicted by the model were as follows: the E/S was 3390 U/g, the solid–liquid ratio was 1:20, the hydrolysis temperature and time were 46 °C and 3.4 h, respectively, and the pH was 7. Under optimal conditions, the predicted value of the α-glucosidase inhibitory rate was 65.2%. To verify the reliability of the predictive value of the model, three validation tests were carried out under optimum conditions, and the average α-glucosidase inhibitory rate of hydrolysate was 64.6 ± 1.2%. The result indicates that the optimized parameters of the model were accurate and reliable.

### 3.8. Separation of Peptides by Ultrafiltration

The molecular weight of the peptide is closely related to the α-glucosidase inhibitory activity. As shown in Figure 3, four fractions were obtained by ultrafiltration from shiitake mushroom protein hydrolysates. The α-glucosidase inhibitory rate of UF-IV was 65.0 ± 3.1%, which was higher than that of the other three fractions. The results indicate that the α-glucosidase inhibitory activity increased with the decrease in molecular weight, which is consistent with the results of previous reports [19,27,37]. Therefore, fraction UF-IV was selected for further purification.

### 3.9. Separation of Peptides by Gel Filtration

Peptides are frequently purified by gel filtration. The UF-IV fraction was further separated by a gel filtration column (Sephadex G-10). The results are presented in Figure 4. A total of eleven peaks were detected and named as F1–F11. The 11 fractions were collected to study their α-glucosidase inhibitory activity. Fraction F5 exhibited the strongest α-glucosidase inhibition rate. The inhibition rate of F5 was 61.3 ± 0.69%.

### 3.10. Identification and Screening of α-Glucosidase Inhibitory Peptides

The sequences of fraction F5 were determined by HPLC–MS/MS with de novo sequencing. Fourteen peptides were identified from fraction F5 (score > 80). The molecular weight of the fourteen peptides ranged from 699 Da to 829 Da (Table 4).

The binding energy and binding sites of peptides to receptor proteins are related to their biological activity. To further screen active peptides from these 14 peptides, the peptides docking with the crystal structure of α-glucosidase (2QMJ) were performed using MOE software. Peptides with binding energy below −14 as well as more than four binding bonds were selected, as shown in Table 5, which were EGEPKLP, KDDLRSP, TPELKL, and LDYGKL (Figure 5). The α-glucosidase inhibition activity of the four peptides was further investigated. EGEPKLP, KDDLRSP, TPELKL, and LDYGKL showed an effective inhibition activity, and their IC50 value was 499 ± 39 μmol/L, 550 ± 37 μmol/L, 452 ± 36 μmol/L, and 696 ± 39 μmol/L, respectively.

### 3.11. Molecular Mechanisms of the Active Peptides

The α-glucosidase inhibition activity of these peptides was obtained by molecular binding with residues in the active center. Figure 5 showed that EGEPKLP formed four hydrogen bonds at Lys534, Leu386, ser521, and Ser288 (Figure 6A). KDDLRSP formed five hydrogen bonds at Lys534, Lys513, His645, Val779, and Asp777 of 2QMJ (Figure 6B). TPELKL formed five hydrogen bonds at Lys534, Lys513, and Glu114 and an arene–cation interaction with His115 residue of 2QMJ (Figure 6C). LDYGKL formed five hydrogen bonds at Lys776, Lys513, and Asp777 of 2QMJ (Figure 6D).

## 4. Discussion

DM is a typical metabolic disease with chronic complications [4,5,6]. With the increasing incidence of diabetes, the controls of blood glucose and therapeutic complications are faced with severe challenges. Intervention with alpha-glucosidase inhibitors can slow the conversion rate of ingested carbohydrates to glucose, thereby preventing glucose from entering the systemic circulation. Compared with the regulatory mechanism of promoting insulin secretion (biguanides, sulfonylureas, DPP IV inhibitors, GLP-1 agonists, etc.), this intervention pathway does not involve islet cells and can relieve the pressure of insulin metabolism to a certain extent and improve the function of damaged islet beta cells. The α-glucosidase in the small intestine plays an important role in the digestion of carbohydrate [19]. Blocking the enzyme with α-glucosidase inhibitors in the digestive tract will limit the digestion of carbohydrates and reduce postprandial hyperglycemia [17,23].

Recently, many studies have confirmed that α-glucosidase inhibitory peptides prepared from food protein have excellent α-glucosidase inhibitory potential [2]. With their higher bioavailability and lower side effects, the study of α-glucosidase inhibitory peptides is important. Different proteases have unique restriction sites, and thus biological peptides with different amino acid compositions can be produced by different proteases. Ren et al. [38] found that the hydrolysates released by amylopsin, pancrelipase, alcalase, papain, and trypsin from hemp seed protein displayed different α-glucosidase inhibition activity. The hydrolysates prepared by alcalase showed a high α-glucosidase inhibition rate. In this research, the α-glucosidase inhibition rates of the hydrolysates obtained by acid protease, alkaline proteinase, neutral proteinase, flavor proteinase, papain, and pepsin were investigated. The neutral proteinase hydrolysate showed the maximum α-glucosidase inhibition activity. This indicates that the newer short peptides with enhanced α-glucosidase inhibitory activity had been generated in the hydrolysates by neutral protease.

Ultrafiltration and gel chromatography can be used to further isolate and purify active peptides from the hydrolysates. In this study, low molecular weight peptides (MW < 1 k Da) had stronger α-glucosidase inhibition activity than other fractions (>5 k Da, 3–5 k Da, 1–3 k Da), which is consistent with the results of previous reports [27,37]. Wang et al. [27] reported that the hydrolysate from ginkgo biloba seed protein was fractionated into five fractions (<1 k Da, 1–3 k Da, 3–5 k Da, 5–10 k Da, >10 k Da), and the fraction with the MW < 1 k Da showed the highest α-glucosidase inhibition activity. Liu et al. [37] found that the peptide fraction (<1 kDa) of wheat germ had better α-glucosidase inhibition activity than other factions. This may be explained by the fact that the active site on the amino acid residues in the low molecular weight peptides can be exposed to the outside and increase the possibility of interacting with α-glucosidase.

The amino acid composition, molecular weight, and chain length of the peptides influence biological activity of α-glucosidase inhibitory peptides [39,40,41]. The majority of peptides with α-glucosidase inhibition activity have a short sequence of less than 10 residues [39,40,42]. The amino acid sequences of fraction with the highest activity were determined by HPLC–MS/MS, and 14 peptide sequences (score > 80) were obtained. The molecular weight of these peptides ranged from 699 to 829 Da and the sequence lengths were 6 to 7 amino acids residues. According to previous reports, the short peptide is beneficial to reducing the free energy of peptide-enzyme binding, and improves the inhibitory effect [40]. The identified peptides were further docked to the crystal structure of α-glucosidase. The peptides with lower energy scores and more binding sites may be proposed as active peptides in shiitake mushroom protein hydrolysates. Among these 14 peptides, EGEPKLP, KDDLRSP, TPELKL, and LDYGKL showed lower energy scores and outstanding binding ability to α-glucosidase.

The IC50 values of EGEPKLP, KDDLRSP, TPELKL, and LDYGKL for α-glucosidase inhibitory activity ranged from 452 ± 36 μmol/L to 696 ± 39 μmol/L. In previous studies, the IC50 value of RVPSLM with α-glucosidase inhibitory activity identified from egg white protein was 23.07 μmol/L [4]. The IC50 of α-glucosidase inhibitory peptide KLPGF from albumin was 59.5 ± 5.7 μmol/L [25]. The four peptides (YLGYLEQLLR, TKVIPYVRYL, RNAVPITPTLNR, FALPQYLK) from Binglangjiang buffalo casein had the inhibitory activity of α-glucosidase and their IC50 values were 470 μmol/L, 498 μmol/L, 504 μmol/L, and 543 μmol/L, respectively [10]. LSMSFPPF, MPGPPSD, and VPKIPPP identified from ginkgo biloba seed protein exhibited the α-glucosidase inhibitory activity with the IC50 values of 454.33 ± 32.45 μmol/L, 943.82 ± 73.10 μmol/L, and 1446.81 ± 66.98 μmol/L, respectively [27]. The IC50 values of SPGAGKG and GLAR from germinated chickpea protein were 1.8 mg/mL and 8.7 mg/mL, respectively [43]. Four α-glucosidase inhibitory peptides were identified from dark tea protein with the IC50 values from 0.04 mg/mL to 1.03 mg/mL [44]. Compared with the previous reports, the activity of EGEPKLP, KDDLRSP, TPELKL, and LDYGKL was of a medium level.

The peptides with more hydrophobic amino acids had better inhibitory activity of α-glucosidase [38,45]. In previous reports, hydrophobic amino acids Leu and Pro were commonly found in the sequence of α-glucosidase inhibition peptides, which contributed greatly to the inhibitory activity, such as LR, PFP, PLMLP, KLPGF, RVPSLM, WLRL, SWLRL and LLPLPVLK [4,16,38,45,46]. Basic amino acids (Lys or Arg) in peptides were also important to α-glucosidase inhibition activity, especially Lys or Arg at the N-terminus [40,47]. The amino acid analysis of the four peptides showed that the proportion of Pro and Leu in EGEPKLP, TPELKL, and LDYGKL was 42.9%, 50.0%, and 33.3%, respectively. Although the proportion of proline and leucine is only 28.6% in KDDLRSP, Lys at the N-terminus may contribute to the activity.

Further study of the interactions between the peptide and α-glucosidase showed that EGEPKLP interacted with the 2QMJ through four hydrogen bonds. KDDLRSP and LDYGKL formed five hydrogen bonds with 2QMJ. TPELKL formed five hydrogen bonds and an arene–cation interaction with 2QMJ. The α-glucosidase inhibition activity is mainly dependent on hydrogen bonding, which agrees with the results of previous research [27,48].

## 5. Conclusions

In this study, the optimal conditions for preparation of α-glucosidase inhibitory peptide from shiitake mushroom were established. Four new α-glucosidase inhibitory peptides (EGEPKLP, KDDLRSP, TPELKL, and LDYGKL) were identified from shiitake mushroom protein hydrolysates using LC-MS/MS and virtual screening. TPELKL exhibited a higher α-glucosidase inhibitory activity than EGEPKLP, KDDLRSP, and LDYGKL. The molecular docking results demonstrated that the hydrogen bond and arene–cation bond were the two major interactions between the four peptides and 2QMJ. The hydrogen bond was crucial to the α-glucosidase inhibition activity. The results suggest that shiitake mushroom may be a reliable source of α-glucosidase inhibitory peptide and provide a feasible strategy for further utilization of shiitake mushroom protein. To improve the possibility of using these peptides as supplements, future studies are required, including examination of the effects of the peptides on other enzymes linked to starch digestion, the antidiabetic mechanism of these peptides, and their safety and stability in vivo.

## Figures and Tables

**Figure 1 foods-12-02534-f001:**
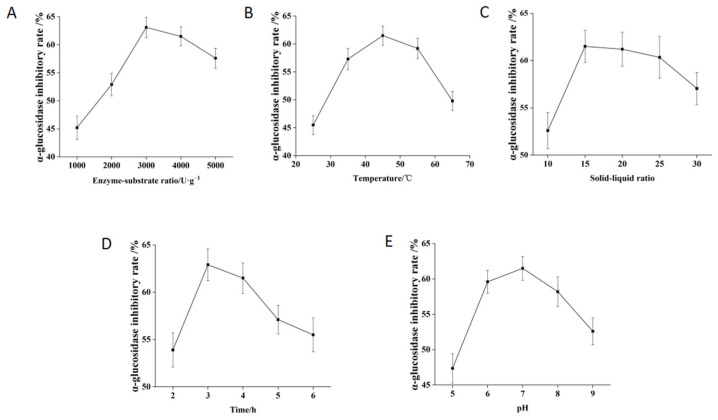
Effects of E/S (**A**), temperature (**B**), solid–liquid ratio (**C**), time (**D**), and pH (**E**) on α-glucosidase inhibition activity.

**Figure 2 foods-12-02534-f002:**
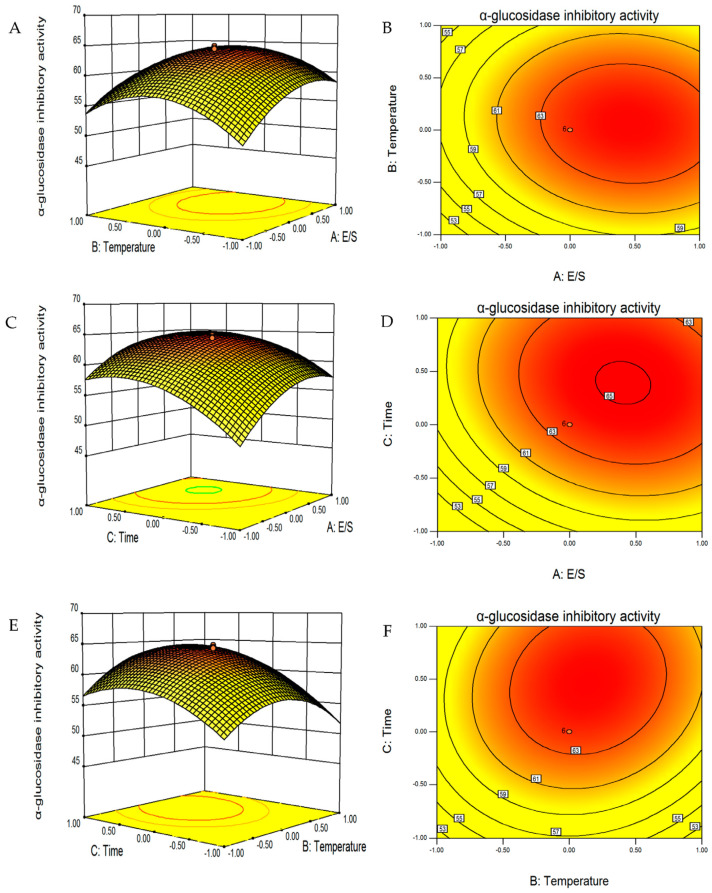
Interaction diagram between factors in the response surface. E/S and temperature (**A**,**B**), E/S and time (**C**,**D**), and temperature and time (**E**,**F**).

**Figure 3 foods-12-02534-f003:**
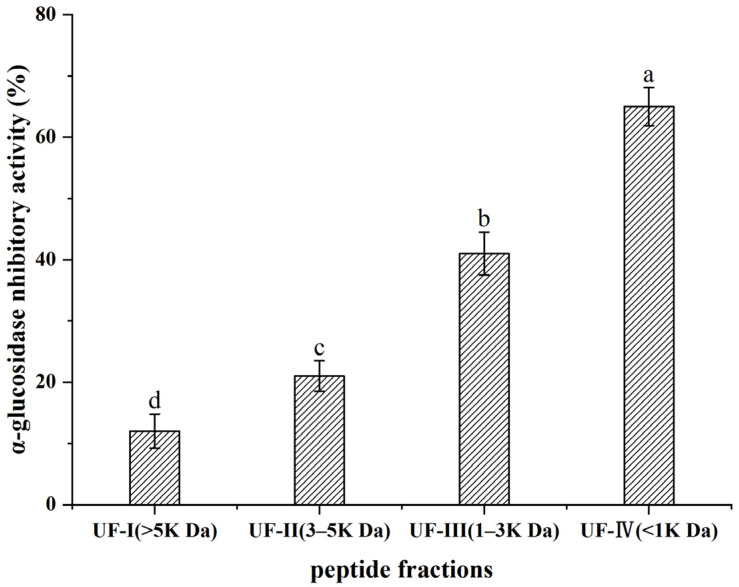
Results of activity of hydrolysate after ultrafiltration. Significant differences are indicated by different letters (*p* < 0.05).

**Figure 4 foods-12-02534-f004:**
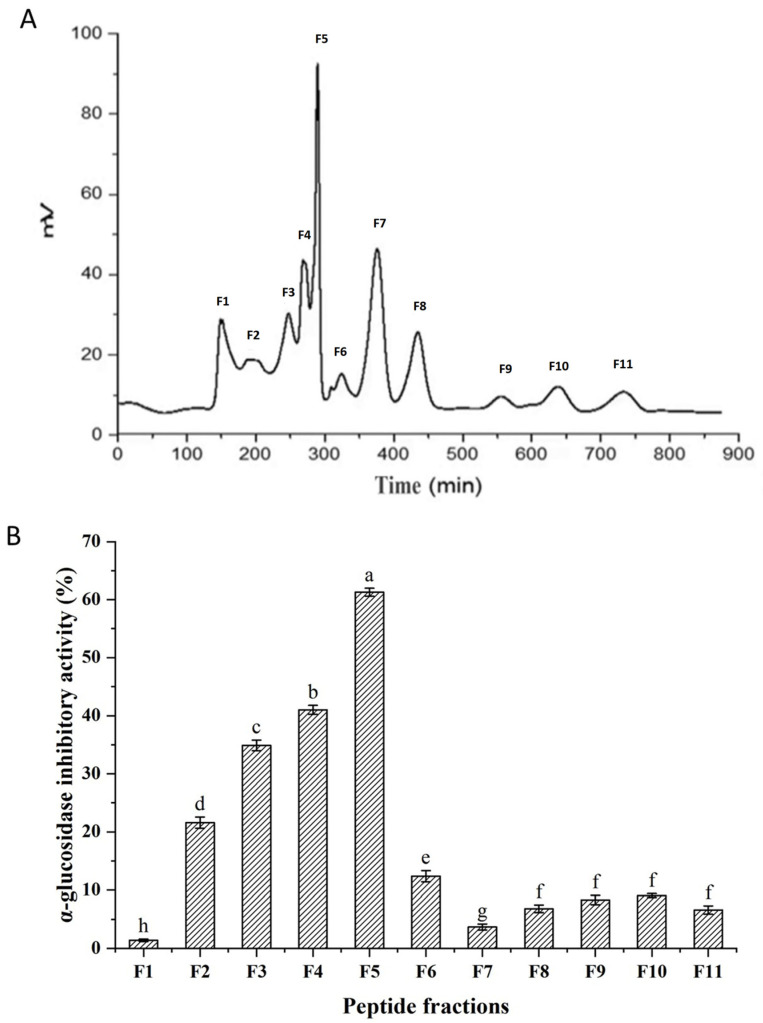
Separation of fraction UF-IV by Sephadex G-10 gel filtration column chromatography. (**A**) Elution profile of the fraction UF-IV; (**B**) α-glucosidase inhibition activity of the fractions separated from UF-IV. Significant differences are indicated by different letters (*p* < 0.05).

**Figure 5 foods-12-02534-f005:**
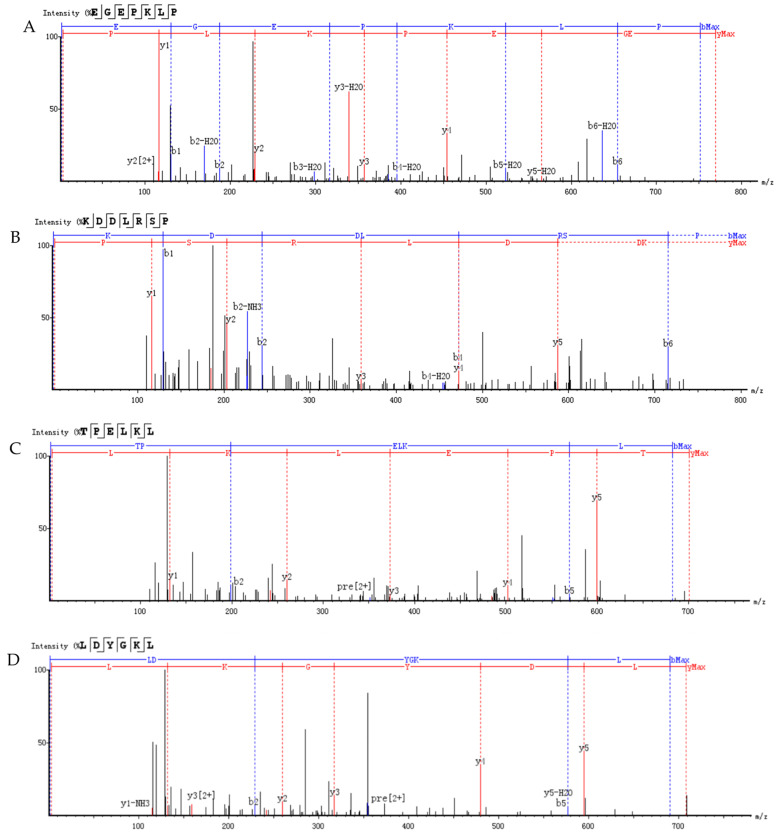
LC-MS-MS chromatogram of F-5. (**A**) EGEPKLP; (**B**) KDDLRSP; (**C**) TPELKL; (**D**) LDYGKL.

**Figure 6 foods-12-02534-f006:**
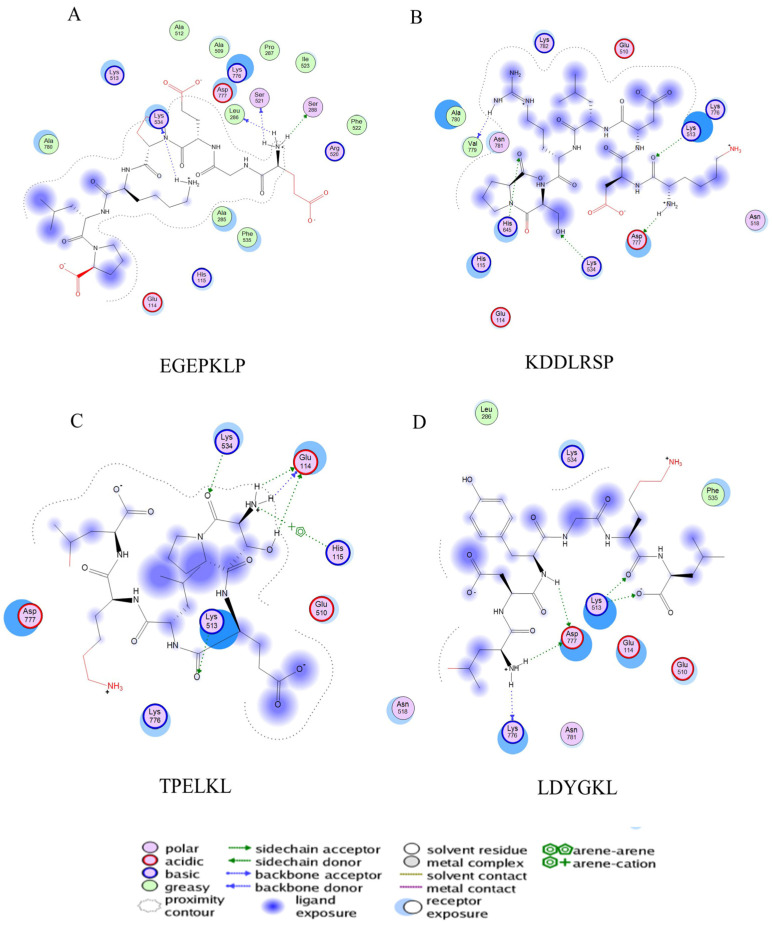
Molecular docking results of peptides. (**A**) EGEPKLP; (**B**) KDDLRSP; (**C**) TPELKL; (**D**) LDYGKL.

**Table 1 foods-12-02534-t001:** Parameters and levels of response surface analysis.

Parameters.	Level				
	−1.68	−1	0	1	1.68
E/S / U/g (A)	1320	2000	3000	4000	4680
Temperature/°C (B)	28.2	35	45	55	61.8
Time/h (C)	1.32	2.00	3.00	4.00	4.68

**Table 2 foods-12-02534-t002:** The experimental results of CCD.

Number	A (E/S u/g)	B (Temperature/°C)	C (Time/h)	Inhibitory Rate/%
1	1.68	0	0	58.2
2	0	0	0	64.3
3	−1	1	−1	43.5
4	0	0	−1.68	48.5
5	−1	−1	−1	43.8
6	1	−1	−1	52.8
7	−1.68	0	0	49.5
8	0	0	0	63.3
9	−1	1	1	55.9
10	0	0	0	64.8
11	1	−1	1	57.3
12	0	0	0	63.1
13	0	0	0	64.5
14	0	0	0	63.2
15	1	1	−1	52.3
16	0	0	1.68	57.7
17	0	−1.68	0	48.5
18	−1	−1	1	49.8
19	0	1.68	0	51.3
20	1	1	1	59.2

**Table 3 foods-12-02534-t003:** Variance analysis results of CCD design.

Source	Squares	df	Square	Value	Prob > F
Model	922.97	9	102.55	89.15	<0.0001
A-E/S	136.85	1	136.85	118.97	<0.0001
B-Temperature	10.38	1	10.38	9.03	0.0132
C-Time	150.08	1	150.08	130.46	<0.0001
AB	2.42	1	2.42	2.1	0.1776
AC	6.12	1	6.12	5.32	0.0437
BC	9.68	1	9.68	8.41	0.0158
A^2^	175.52	1	175.52	152.58	<0.0001
B^2^	344.1	1	344.1	299.13	<0.0001
C^2^	203.2	1	203.2	176.65	<0.0001
Residual	11.5	10	1.15		
Lack of Fit	8.69	5	1.74	3.09	0.1207
Pure Error	2.81	5	0.56		
Cor Total	934.48	19			
Model	922.97	9	102.55	89.15	<0.0001
A-E/S	136.85	1	136.85	118.97	<0.0001
B-Temperature	10.38	1	10.38	9.03	0.0132
C-Time	150.08	1	150.08	130.46	<0.0001
AB	2.42	1	2.42	2.1	0.1776
AC	6.12	1	6.12	5.32	0.0437

**Table 4 foods-12-02534-t004:** Peptide sequences identified from fraction F5 (score > 80).

Sequence	Mass	*m*/*z*	*m*/*z* Error (ppm)	RT (min)	Length	Score
DVFAHF	734.3387	368.1776	2.5	20.7	6	82
KDDLRSP	829.4293	415.7215	−1.1	11.98	7	83
EDLRLP	741.402	371.7031	−14.1	15.63	6	83
LLAKFE	719.4218	360.7188	1.8	14.72	6	83
EPLEPK	711.3802	356.6971	−0.7	11.21	6	86
LQHLPL	719.433	360.7242	1.3	15.72	6	87
VLSRKL	714.4752	358.2401	−13.2	11.41	6	88
EGEPKLP	768.4017	385.208	−0.4	12.45	7	88
LDYGKL	707.3854	354.699	−2.6	13.24	6	88
TPELKL	699.4167	350.7144	−3.6	11.89	6	88
SPDEPKL	784.3967	393.2057	0.4	11.35	7	90
EEPLPQ	711.3439	356.679	−0.5	12.75	6	92
VVELLK	699.4531	350.7336	−0.6	13.53	6	96
DPEKFP	731.3489	366.6822	1.2	12.79	6	97

**Table 5 foods-12-02534-t005:** The peptides with binding energies below −14.

Sequence	E-Score	Site	Molecular Structural Formula
EGEPKLP	−15.42	Lys534, Leu286, Ser521, Ser288	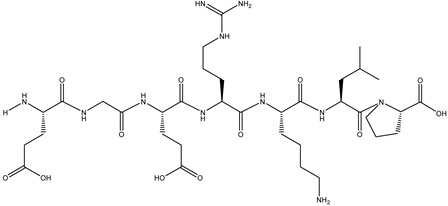
KDDLRSP	−15.83	Lys534, Lys513, His645, Val779, Asp777	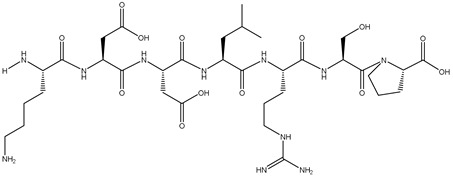
TPELKL	−14.25	Lys534, Lys513, His115, Glu114 (3 binding bonds)	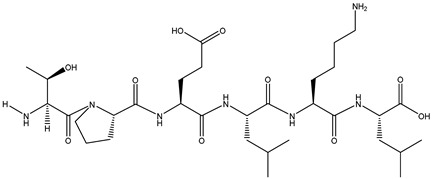
LDYGKL	−16.31	Lys776, Lys513 (2 binding bonds), Asp777 (2 binding bonds)	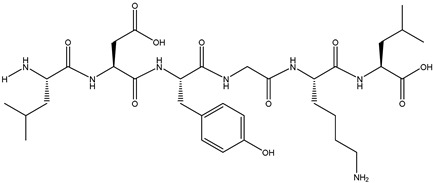

## Data Availability

The datasets generated for this study are available on request from the corresponding author.

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
