# Peer review of "Preparation and Identification of Peptides with α-Glucosidase Inhibitory Activity from Shiitake Mushroom (*Lentinus edodes*) Protein"

_foods, 2023, doi:10.3390/foods12132534_

Round 1

Author Response

The article deals with the Preparation and Identification of Peptides with α-glucosidase Inhibitory Activity from Shiitake Mushroom (Lentinus edodes) Protein. Congratulations on the search. I hope you understand that my purpose is to contribute to the improvement of the manuscript.

It is an interesting and relevant research; however, a general revision of the manuscript is needed in the items material and methods and results and discussion.

The introduction is in line and very well written.

Point 1: Abstract: It is necessary to include the results of the hydrolysis optimization. The sentence "Therefore the α-glucosidase peptides from the shiitake mushroom protein hydrolysate could be used as functional foods and drugs for hyperglycemia treatment" needs to be revised. For this statement to be made, it is necessary that other enzymes linked to starch digestion be studied. Redo the conclusion based on the studied enzyme.

Response 1: We are very thankful for reviewer’s suggestions, and we have add the results of the hydrolysis optimization in the abstract (line 18 to 22), and revised the sentence "Therefore the α-glucosidase peptides from the shiitake mushroom protein hydrolysate could be used as functional foods and drugs for hyperglycemia treatment" into as follow “The results indicate the potential of using the peptides from shiitake mushroom protein as functional food with α-glucosidase inhibition activity”.

Point 2: In item 2 - Material and Methods:

In general, it is necessary to include more information than we detail procedures adopted. In this context, it is necessary to make a careful review of item 2. Also, I suggest that items 2.2, 2.4 and 2.5 are approximate, since they are complementary. They can even be combined into one item. It is at the discretion of the authors; however, it is necessary that the steps are subsequent for a better understanding.

Response :We have combined items 2.2, 2.4 and 2.5 into 2.2.

Item 2.1:

  • Include enzymatic activity data for all commercial enzymes used;
  • Include all reagents used in the study and their purity;

Response: We have added the enzymatic activity data for all enzymes and all reagents used in the study and their purity in items 2.1(line 84 to 91, marked it in yellow).

Item 2.2: In general, it is necessary to better detail the steps used in this item.

  • Line 82: include the mass-volume ratio of the 1:20 ratio... example: is 1g in 20 mL? pH was adjusted with what? in the rest of item 2 it is necessary to make this same adjustment (example line 89);
  • Line 83: what is the frequency of the sonicator?
  • ine 86: in what proportion was the hydrolysate dissolved in water? what type of water? (distilled, deionized, ultrapure...);
  • Line 87: how much, in units of enzymatic activity, was included from the protease? if this is one of the variables in item 2.5, mention it here - according to item 2.5;
  • Line 88: would it be rpm/min?
  • Line 90: what is the centrifugation condition?

Response: We have detailed the steps in item 2.2 (now 2.2.1, line 101-115) according to the suggestion.

  • Line 92: The concentration of alpha-glucosidase equals how many units of enzyme activity? include;

Response: We have added the enzyme activity in item 2.3 (line 133) according to the suggestion.

  • Detail the experimental conditions of item 2.6.2;

Response: We have detailed the experimental conditions of item 2.6.2(now 2.4.2, line 146-150).

  • Detail the experimental conditions of item 2.7;

Response: We have detailed the experimental conditions of item 2.7 (now 2.5, line 155-166).

  • Put the molecular coupling in a single item and detail the conditions used: what type of doking? what is the population used? how many repetitions per coupling? How did you get the structures of the peptides? all details and software/platforms must be described.

Response: We have put the molecular coupling in item 2.6 and detailed the conditions in line 167-178.

  • Item 2.8: How were the peptides synthesized? To describe.

Response: We have added the method of peptides synthesis in 2.7 (line 180-190).

  • Item 2.9: what statistical tests are used? what is the confidence level applied? how many repetitions per test were performed?

Response: We have detailed the method of statistical tests in 2.8 (line 192-194).

Point 3: In Conclusion: improve considering all the aspects studied in the research. What does the study contribute to the specific area? include.

Response 3: We revised the conclusion and added the the contribution in shiitake mushroom utilization of the research, and the improve considering in future studies. see line 440-445.

Reviewer 2 Report

In the presented manuscript, the authors performed the optimization of enzymatic hydrolysis of proteins from shiitake mushrooms to identify peptides possessing α-glucosidase inhibition activity. The search for new peptides with α-glucosidase inhibition activity in food products is important because as documented inhibition ofα-glucosidase activity has grown up to be the effective method for T2DM treatment because of its role in delaying the hydrolysis of carbohydrates to inhibit postprandial hyperglycemia. In my opinion, the level of performed research is significant and valuable. The manuscript is well planned but analyzing the data included in the work, I have the impression that the authors treated the part concerning the optimization of hydrolysis conditions and the study of biological activity in much more detail, while they paid much less attention to the analysis of these new peptides showing this activity. To analyze the mixture of active peptides, the authors used the LC-MS and LC-MS/MS methods. This is a key and very important method without which such an analysis would be impossible. However, the manuscript does not include any mass spectrum, any fragmentation spectrum, any technical details regarding the type of mass spectrometer used, the ion source, the solvents used for separation, the method used for separation (gradient or isocratic), etc. The experiments prepared in this way are impossible to repeat and unacceptable.

The work should be supplemented with the following point:

1. Detailed description of the equipment used in the experiments.

2. Detailed description of HPLC-MS and MS/MS measurement conditions.

3. Example ESI-MS and ESI-MS/MS spectra.

4. If peptides were purchased to check their activity, was the retention time of at least one compared with the retention time of the same peptide identified in hydrolysate to further confirm the identity of this peptide?

5. More details concerning the molecular docking – what program was used, the parameters etc.

6. In Table 4, please indicate with what error the m/z values were measured.

7. I strongly suggest improving the editing side of the work - lots of missing spaces, sometimes italics used sometimes not. It's too much to point out.

Author Response

Point 1: Detailed description of the equipment used in the experiments.

Response 1: We have detailed description of the equipment used in the experiments in item 2.1 (line 92-99).

Point 2: Detailed description of HPLC-MS and MS/MS measurement conditions.

Response 2: We have detailed the description of HPLC-MS and MS/MS measurement condition in item 2.5 (line 155-166).

Point 3: Example ESI-MS and ESI-MS/MS spectra.

Response 3: We have added the spectra in Figure 5.

Point 4: If peptides were purchased to check their activity, was the retention time of at least one compared with the retention time of the same peptide identified in hydrolysate to further confirm the identity of this peptide?

Response 4: We think this is an excellent suggestion to further confirm the identity of the peptides. In this manuscript, we identified these peptides by LC-MS-MS and combined with the search of mushroom protein database to further confirm them. The synthetic peptides were confirmed by China Peptides Co., LTD before shipping to us. Because the peptides synthesis need three weeks, we did not test them at the same time. In the future research, we will use this method in our study. Thanks for your suggestion.

Point 5: More details concerning the molecular docking – what program was used, the parameters etc.

Response 5: We have put the molecular docking in item 2.6 and detailed the program of molecular docking in line 167-178.

Point 6: In Table 4, please indicate with what error the m/z values were measured.

Response 6: We have added the error of the m/z values in Table 4.

Point 7: I strongly suggest improving the editing side of the work - lots of missing spaces, sometimes italics used sometimes not. It's too much to point out.

Response 7: We tried our best to improve editing side of the manuscript. here we did not list the changes, but marked them in yellow in the revised manuscript.
